

# Comparative transcriptome and hormone analyses of roots in apple among three rootstocks with different rooting abilities

Zhongyong Li[1,*], Yang Cao[1,*], Jie Zhu[1,2], Jin Liu[1], Feng Li[1], Shasha Zhou[1], Xueying Zhang[1], Jizhong Xu[1] and Bowen Liang[1]

[1] Hebei Agricultrual University, Baoding, China
[2] Shijiazhuang Institute of Pomology, Hebei Academy of Agriculture and Forestry Sciences, Shijiazhuang, China
[*] These authors contributed equally to this work.

## ABSTRACT

**Background**. Root plays an important role in the growth and development of fruit trees; however, the molecular mechanisms behind the differences among rootstock varie-ties remain unclear.

**Methods**. This study examined the effects of different rootstocks on root structure and the endogenous hormone content of 1-year old apple seedlings in combinations of Tianhong 2 (T2)/*Malus robusta* (HT), T2/G935, and T2/Jizhen 2 (J2).

**Results**. The results showed that the T2/HT treatment had greater root length, surface area, volume, average diameter, tips and forks, followed by G935 and J2. In T2/HT leaves and roots, the indole-3-acetic acid (IAA) and gibberellins (GA$_3$) levels were highest, and the abscisic acid (ABA) levels were the lowest. A root transcriptome analysis detected 10,064, 10,511, and 8,719 differentially expressed genes in T2/HT *vs.* T2/G935, T2/HT *vs.* T2/J2, and T2/J2 *vs.* T2/G935, respectively. The analysis of Gene Ontology (GO) terms indicated a significant enrichment in biological processes, cellular components, and molecular functions. Kyoto Encyclopedia of Genes and Genomes (KEGG) analysis showed that plant hormone signaling, MAPK signaling pathway–plant, and plant–pathogen interaction played important roles in differences in the rooting ability of different rootstocks. In addition, some key differential genes were associated with root growth and development and were involved in these metabolic pathways. This study is important for enriching theoretical studies of fruit tree roots.

# INTRODUCTION

The apple (*Malus domestica*) is an important fruit tree in northern China (*Xu et al., 2021*). Its cultivation area and yield are increasing and it has significant economic value (*Zhou et al., 2021*). Plant roots are vital for anchoring plants in the soil, supporting plant development, water and nutrients, and adapting to the environment (*Zluhan-Martínez et al., 2021*). The root system expands by developing lateral roots and root hairs, which are specialized structures produced by epidermal cells that increase the root surface area to facilitate the absorption of nutrients and water (*Serrano et al., 2023*). During root development, various

Corresponding author
Bowen Liang, lbwnwsuaf@126.com

coordinated activities such as cell division, elongation, differentiation, and the growth of root hairs and lateral roots contribute significantly (*Salvi et al., 2020*). A robust root system is crucial for apple plants to support growth and development in challenging conditions.

Grafting is a vital practice in apple cultivation that enhances productivity, controls growth, and improves fruit quality. The interactions between scion and rootstock are complex and significantly influence the overall success of the grafted tree (*Chai et al., 2022*). The rootstock is fundamental in apple cultivation (*Adams et al., 2018*; *Li et al., 2022a*). The use of rootstock in modern apple production has multiple benefits, including improved productivity, controlled tree growth and yield, enhanced resource use efficiency, and equipping the scion with necessary traits for withstanding internal and environmental stresses (*Ji et al., 2023*). The root structure differs among various apple rootstocks (*Liang et al., 2022*). The differences apple rootstock root morphology primarily lie in characteristics such as length, surface, average diameter, tips, forks, and volume (*Liu et al., 2019*). The root architecture of various apple rootstock varieties varies significantly, and these differences may have distinct effects on the growth and development of apple plants (*Zhou et al., 2022*). Furthermore, the rate of root growth varies significantly among different apple rootstocks. In general, rootstock varieties with rapid growth are more adaptable and thrive, which contributes to improved fruit quality (*Zhou et al., 2022*). The growth rate is closely related to the genetic characteristics of stock varieties. The yield and quality of apples can be enhanced by breeding cultivars with rapid growth rates (*Butkeviciute, Janulis & Kviklys, 2022*; *Zhou et al., 2022*). However, some cultivars may struggle to root due to genotypic differences in their rooting ability. Researching the mechanisms of varietal differences in rootstock root formation is beneficial for selecting superior varieties, improving fruit quality, and adapting trees to diverse environments.

Plant hormones play an important role in regulating root formation (*Wen et al., 2023*). The formation of apple rootstock is a complex process primarily influenced by auxin signaling pathways. In addition to auxin, multiple hormone signaling pathways interact and contribute to the regulation of root formation in apple rootstocks (*Li et al., 2022a*). The levels of indole-3-acetic acid (IAA) and gibberellins ($GA_3$) are higher during the peak growth period of apple roots (*Liang et al., 2020*). Greater accumulation of abscisic acid (ABA) and the upregulation of related genes inhibit root growth (*Li et al., 2022b*). Under alkali stress, elevated levels of IAA and ABA in the roots, with the upregulation of relevant genes, enhance root growth and increase tolerance of apple rootstocks. Transcriptome analyses have identified many differentially expressed genes (DEGs) involved in hormone signaling pathways in apples (*Liu et al., 2019*). For the past few years, transcriptomics has become popular for exploring signaling pathways during root formation (*Fan et al., 2022*). However, there are few comparative studies of the root transcriptomes of different apple rootstocks.

Therefore, this study compared different rootstock/scion combinations (Tianhong 2 (T2)/*Malus robusta* (HT), T2/G935, and T2/Jizhen 2 (J2)) using transcriptome and hormone analyses to reveal the mechanisms by which plant hormones regulate root formation in different apple varieties and provide new ideas for apple rootstock selection.

## MATERIALS & METHODS

### Plant materials and experimental design

Experimental trials were conducted at Hebei Agricultural University, Baoding (38°23′N, 115°38′E), Hebei, China. Three different apple rootstocks were used: HT, G935, and J2. Seedlings were planted in plastic pots (35 × 20 × 22 cm²) containing sand and soil (1:2, v:v). After 1 week of growth, the rootstock was grafted with one-year-old T2. The rootstock was grafted by grafting single-bud technique. The apple trees were grown in a greenhouse under natural temperature, and the pots were rotated once a week to ensure uniform growth. During the growing period, the mean temperature was 26 °C. After grafting, 1/2 Hoagland-Arnon nutrient solution was used for watering, and water was irrigated every 3d thereafter to ensure the growth of young trees. There were 125 grafted apple plants per rootstock–scion combination. The first sampling was performed after 60 days of growth after grafting, denoted as day 0. Shoot length, root architecture, and hormone content were measured on days 0 (June 1st), 30 (July 1st), 60 (August 1st), 90 (September 1st), and 120 (October 1st).

### Determination of root architecture

Destructive sampling was performed every 30 days with three replicates per rootstock–scion combination. The roots were unfolded in water, and root images were obtained using a scanner (Epson Expression 1680; Epson, Sydney, Australia). The surface area, length, volume, average diameter, tips, and forks were measured using the Wanshen LA-S series plant image analyzer system (*Cao et al., 2023*).

### Phytohormone determination

About 200 mg frozen tissues were finely ground into powder. The samples were then extracted using a mixture of isopropanol and formic acid, and subsequently injected into LC-MS/MS 8030 system (Shimadzu, Kyoto, Japan). Phytohormone contents were calculated according to their respective internal standards.

### RNA extraction, illumine sequencing, and transcriptome data analysis

RNA extraction, illumine sequencing, and transcriptome data analysis were performed by Wuhan Metaville Biotechnology (http://www.metware.cn, Wuhan, China). Total RNA was extracted from the roots of three rootstocks using TRIzol reagent (Invitrogen, Carlsbad, CA, USA), three biological replicates were performed for each sample. The cDNA libraries were sequenced using an Illumina HiSeq platform (Illumina, San Diego, CA, USA). For each transcription region, the fragments per kilobase of transcripts per million mapped reads (FPKM) were calculated using RESM software. A false discovery rate (FDR) of 0.05 was used as the threshold $P$-value in tests to evaluate the significance of gene expression differences. A |log2FoldChange| >1 was set as the threshold for significant differential expression. The Gene Ontology (GO) and Kyoto Encyclopedia of Genes and Genomes (KEGG) tools were used to analyze the DEGs.

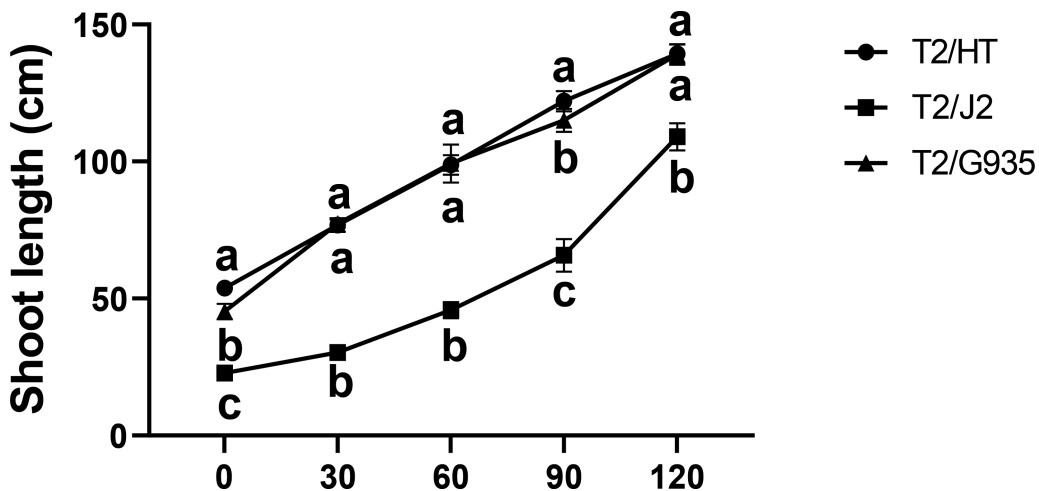

**Figure 1** **Effects of different rootstocks on shoot length of apple trees.** Values labelled with different letters are significantly different ($n = 3$, $p < 0.05$).

### RT-qPCR analysis

RNA extracted from the roots of three different apple rootstocks—HT, G935, and J2—was utilized for reverse transcription quantitative polymerase chain reaction (RT-qPCR) analysis. The RT-qPCR primers are listed in Table S1. The expression *MD00G1000200*, *MD00G1001500*, *MD00G1003500*, and *MD00G1007800* was evaluated. *β-Actin* was used as an internal reference gene. Total RNA was extracted using the M5 Plant RNeasy Complex Mini Kit (Mei5 Biotechnology, Beijing, China), as directed by the manufacturer. Fluorescence quantification using Fast Super EvaGreen qPCR Master Mix (S2008S) (US Everbrite, Suzhou, China). The relative expression of each gene was analyzed using $2^{-\Delta\Delta CT}$.

### Statistical analyses

Differences were identified using SPSS v26.0 (IBM, Armonk, NY, USA). The graphs were visualized using GraphPad Prism v9.0 (GraphPad, San Diego, CA, USA). Significant differences between means were assessed using Tukey's range test, and significance was evaluated at $p < 0.05$.

## RESULTS

### Effects of different rootstocks on the apple tree shoot length

HT, G935, and J2, which have significant differences in rooting ability, were studied. Compared with the shoot length at day 0, the shoot lengths of HT, J2, and G935 were significantly increased by 158.9%, 378.1%, and 207.2% at day 120, respectively. After 120 days of growth, there was no significant difference in shoot length between HT and G935, while that of J2 was significantly greater (Fig. 1).

### Effects of different rootstocks on apple tree root architecture

After 120 days of growth, rootstock HT had the greatest root length, surface area, volume, average diameter, tips, and forks, followed by G935 and J2. Compared with J2, the root

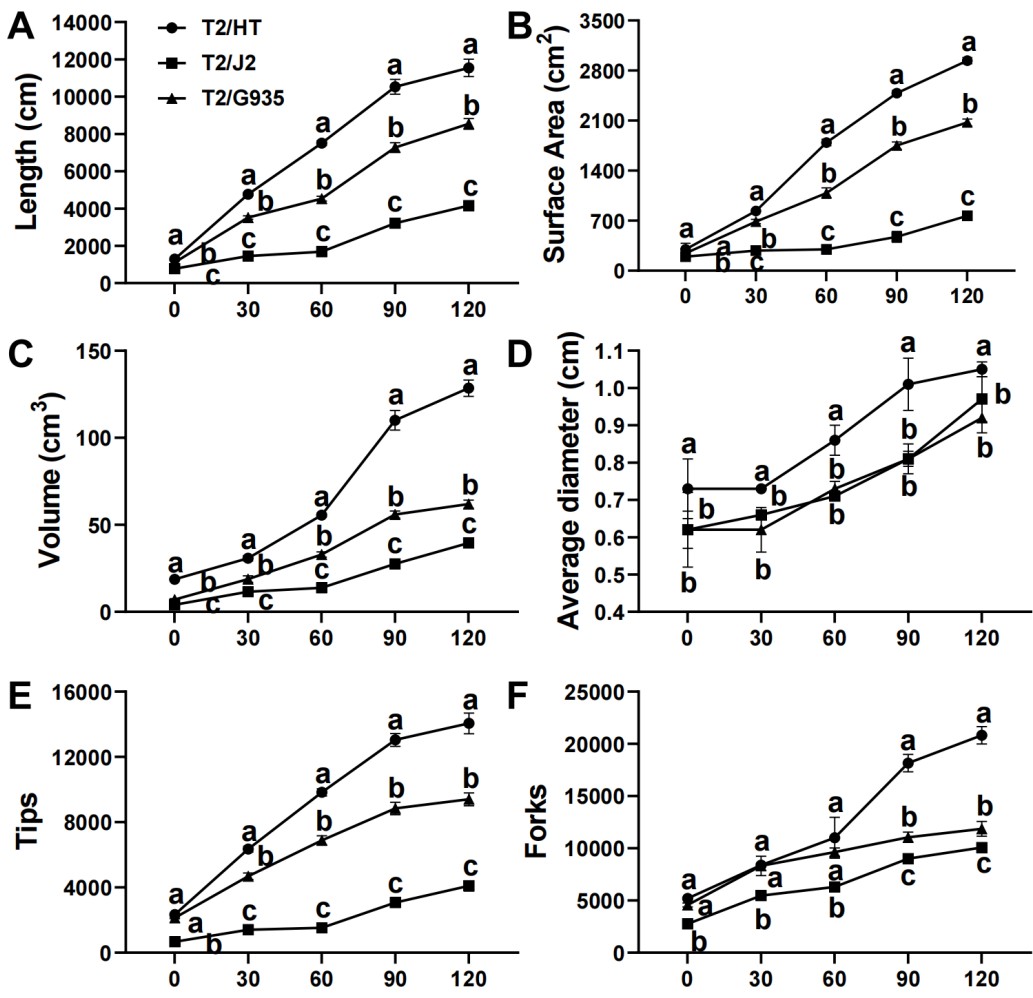

**Figure 2 Effects of different rootstocks on root architecture of apple trees.** (A) The length, (B) surface area, (C) volume, (D) average diameter, (E) tips and (F) forks. Values labelled with different letters are significantly different ($n = 3$, $p < 0.05$).

length, surface area, volume, average diameter, tips, and forks of HT were significantly increased by 177.6%, 282.8%, 225.2%, 8.2%, 243.1%, and 243.1%; the same parameters were changed significantly by 105.4%, 170.2%, 56.7%, −5.2%, 129.5% and 129.4% in G935, and by 35.2%, 41.7%, 107.5%, 14.1%, 49.5%, and 75.6% in HT (Fig. 2).

## Changes in phytohormones in roots and leaves of different rootstocks

The phytohormones ABA, IAA and GA$_3$ differed remarkably among the three rootstocks. Compared with J2, the respective IAA and GA$_3$ contents of HT were significantly increased by 25.4% and 13.9% in leaves, and by 64.6% and 18.6% in roots, while the ABA content of HT was significantly decreased by 9.0% in leaves and by 24.7% in roots. Compared with J2, the IAA and GA$_3$ contents of G935 were significantly increased by 23.0% and 12.1% in leaves and by 38.7% and 8.1% in roots, while the ABA content of G935 was significantly decreased by 8.7% in leaves and 19.5% in roots. Compared with G935, the IAA and GA$_3$

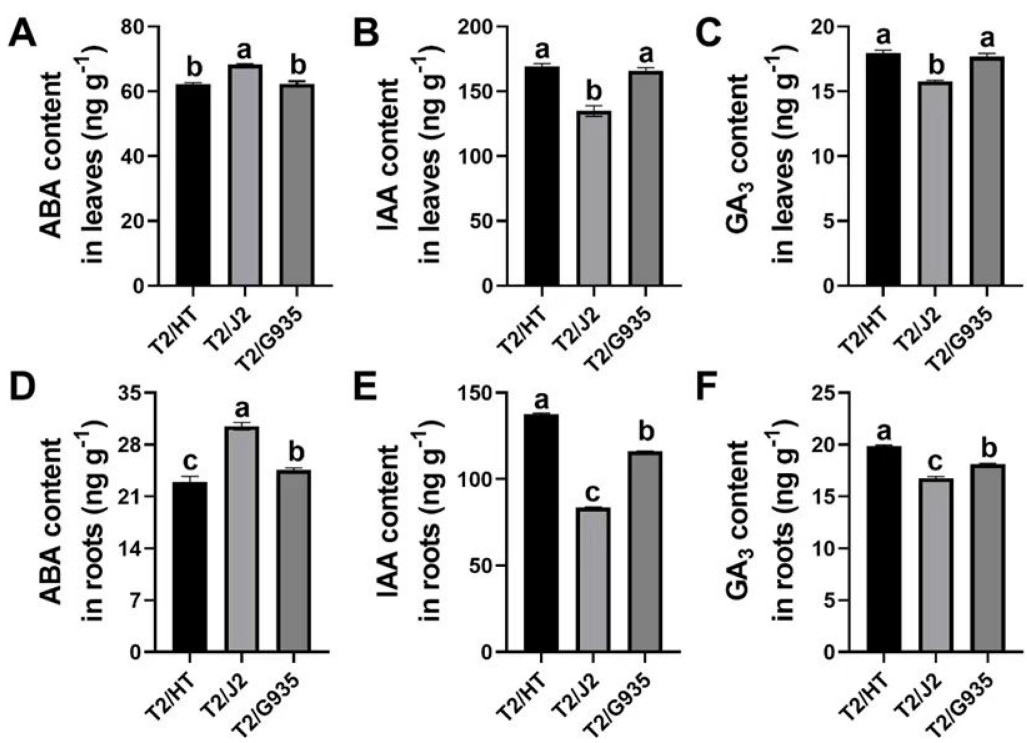

**Figure 3 Effects of different rootstocks on plant hormone of apple trees.** (A, D) ABA content, (B, E) IAA content, (C, F) GA3 content. Values labelled with different letters are significantly different ($n = 3$, $p < 0.05$).

contents of HT were significantly increased by by 18.7% and 9.7% in roots, while the ABA content was significantly decreased by 6.4% in roots (Fig. 3).

## RNA sequencing and data analysis

Nine cDNA libraries from three rootstocks each of HT, J2, and G935 were constructed and sequenced. Three pairs of libraries (T2/HT *vs.* T2/G935, T2/HT *vs.* T2/J2, and T2/J2 *vs.* T2/G935) were prepared and transcriptome analysis was performed on the apple roots (three replicates per treatment). After eliminating low-quality data and reads containing adapters, 100.6 Gb of clean data were generated and 10 Gb of clean data were obtained for each sample. The raw sequencing data have been submitted to the NCBI Sequence Read Archive (SRA) database (PRJNA1066895). The statistical power of this experimental design, calculated in RNASeqPower is 0.89 (T2/HT *vs.* T2/G935), 0.90 (T2/HT *vs.* T2/J2), and 0.86 (T2/J2 *vs.* T2/G935). The Q20 and Q30 of the raw data were over 97% and 93%, respectively, indicating that the quality of the transcriptome data was relatively high (Table 1). The correlation coefficients among the three treatment groups demonstrated the reliability and the appropriateness of sample selection for subsequent analysis (Fig. 4A). Furthermore, principal component analysis (PCA) confirmed distinct expression patterns of gene clusters across the different treatments (Fig. 4B). Further, 1246 overlapping DEGs were detected in all three pairwise comparisons (Fig. 4C). After DESeq2 was used to

**Table 1 Statistical table of sequencing output.**

| Sample | Raw reads | Clean reads | Reads mapped | Q20 (%) | Q30 (%) | GC content (%) |
|---|---|---|---|---|---|---|
| T2/HT-1 | 83,279,986 | 75,858,850 | 65,566,516 (86.43%) | 98.09 | 94.44 | 46.07 |
| T2/HT-2 | 83,481,896 | 76,463,786 | 65,970,240 (86.28%) | 97.94 | 94 | 46 |
| T2/HT-3 | 76,712,006 | 73,126,324 | 63,180,854 (86.40%) | 97.77 | 93.65 | 45.96 |
| T2/J2-1 | 80,904,614 | 75,815,900 | 58,219,877 (76.79%) | 98.13 | 94.53 | 44.9 |
| T2/J2-2 | 83,140,652 | 77,547,910 | 59,462,772 (76.68%) | 98.05 | 94.3 | 44.85 |
| T2/J2-3 | 78,086,586 | 71,938,668 | 54,912,875 (76.33%) | 97.97 | 94.2 | 44.82 |
| T2/G935-1 | 80,241,568 | 75,438,708 | 63,321,897 (83.94%) | 97.99 | 94.14 | 45.77 |
| T2/G935-2 | 73,628,784 | 68,474,140 | 57,384,899 (83.81%) | 98.13 | 94.52 | 45.84 |
| T2/G935-3 | 81,728,660 | 76,147,926 | 63,888,974 (83.90%) | 97.93 | 94.1 | 45.88 |

complete the DEG analysis, the total number of DEGs and numbers of upregulated and downregulated genes in each group were counted. There were 10,064 (5,212 up- and 4,852 downregulated) DEGs in T2/HT *vs.* T2/G935 (Fig. 5A), 10,511 (5,142 up- and 5,369 downregulated) in T2/HT *vs.* T2/J2 (Fig. 5B), and 8,719 (4,218 up- and 4,501 downregulated) in T2/J2 *vs.* T2/G935 (Fig. 5C). Following the retrieval of FPKM values for the genes showing differential regulation, we performed clustering analysis to group genes with comparable expression patterns. Our observations revealed prominent disparities in gene expression when comparing different groups (Fig. 5D).

## Functional annotation and enrichment analysis of DEGs

According to GO categories, these DEGs were categorized into biological processes, cellular components, and molecular functions. GO classification bar charts were plotted by ranking the number of genes on the annotation from largest to smallest (Fig. 6). Comparing T2/HT and T2/G935, 10,064 DEGs were divided into three primary categories (biological processes, cellular components, and molecular functions) and further separated into 43 subgroups (Table S2 and Fig. 6A). For T2/HT *vs.* T2/G935, the top three GO enrichment terms for biological processes were cellular process (GO:0009987), metabolic process (GO:0008152), and response to stimulus (GO:0050896). The top three GO enrichment terms for molecular function were binding (GO:0005488), catalytic activity (GO:0003824), and transcription regulator activity (GO:0140110) and the top GO enrichment term for cell components was cellular anatomical entity (GO:0110165) (Table S2 and Fig. 6A). Comparing T2/HT *vs.* T2/J2, 10,511 DEGs were divided into three primary categories and further separated into 43 subgroups. For T2/HT *vs.* T2/J2, the main GO classifications were the same as for T2/HT *vs.* T2/G935 (Table S3 and Fig. 6B). Comparing T2/J2 and T2/G935, 8719 DEGs were divided into three primary categories and further separated into 44 subgroups. For T2/J2 *vs.* T2/G935, the main GO classifications were the same as for T2/HT *vs.* T2/G935 (Table S4 and Fig. 6C).

To understand the dynamic biological pathways in DEGs among various treatments comprehensively, and identify the key pathways, KEGG pathway enrichment analysis was conducted based on the expression profile. Tables S5, S6, and S7 show the primary differential biological pathways of these DEGs in apple roots. These show annotated

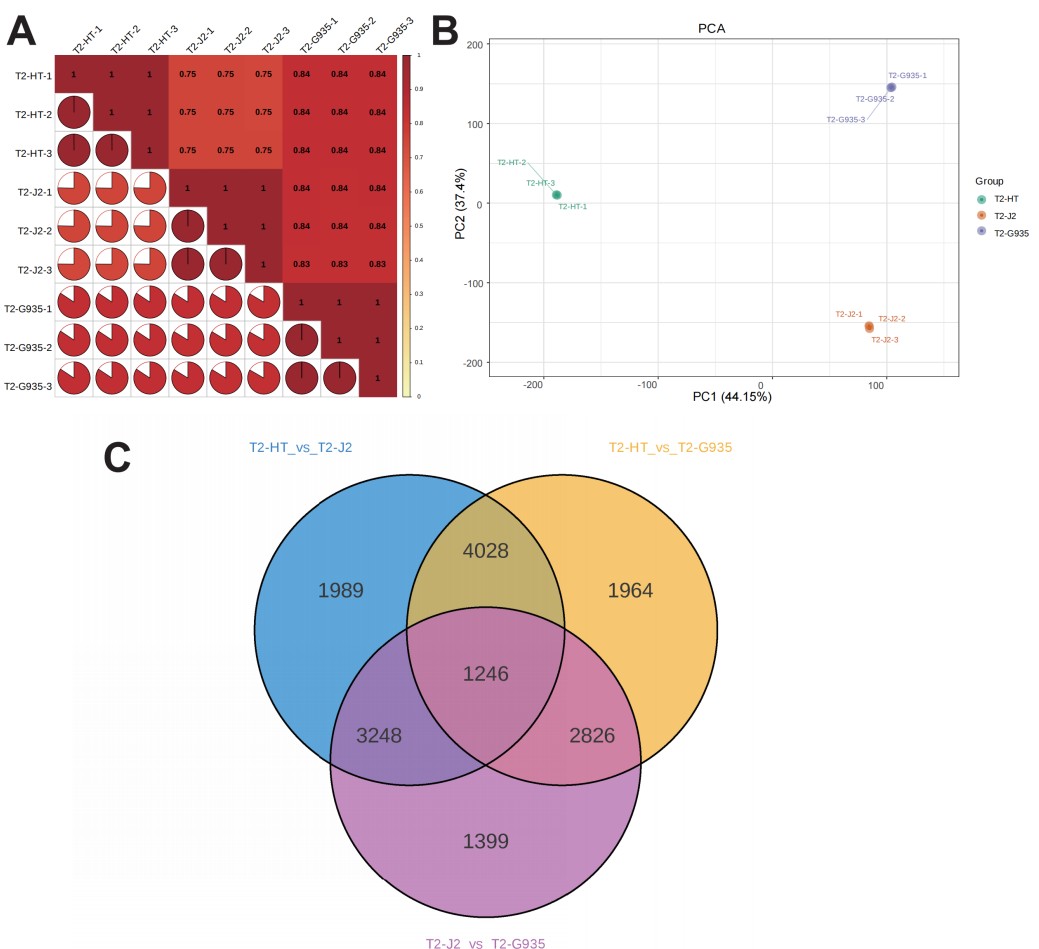

**Figure 4 Correlation heat map, principal component analysis (PCA) score plot, and the number of differential expression genes (DEGs) in transcriptomic profile of apple root samples.** (A) Pearson's correlation analysis of patterns of gene expression. (B) Each point in PCA score plot representing an independent biological replicate. (C) Venn diagram of the number of DEGs with each pairwise comparison.

3438 DEGs to 141 metabolic pathways in the comparison between T2/HT *vs.* T2/G935, 3,641 DEGs to 141 metabolic pathways in the comparison between T2/HT *vs.* T2/J2, and 2,933 DEGs to 141 metabolic pathways in the comparison between T2/J2 *vs.* T2/G935. A scatterplot was used to display the results of the KEGG enrichment analysis (Fig. 7). For T2/HT *vs.* T2/G935, the paths with the most significant enrichment were plant–pathogen interaction, MAPK signaling pathway–plant, flavonoid biosynthesis, plant hormone signal transduction, secondary metabolite biosynthesis, and steroid biosynthesis (Fig. 7A). For T2/HT *vs.* T2/J2, the paths with the most significant enrichment were plant–pathogen interaction, plant hormone signal transduction, secondary metabolite biosynthesis, nitrogen metabolism, MAPK signaling pathway–plant, and flavonoid biosynthesis (Fig. 7B). For T2/HT *vs.* T2/J2, the paths with the most significant enrichment were plant–pathogen interaction and MAPK signaling pathway–plant (Fig. 7C). In addition, some key differential genes related to root growth and development were screened out,

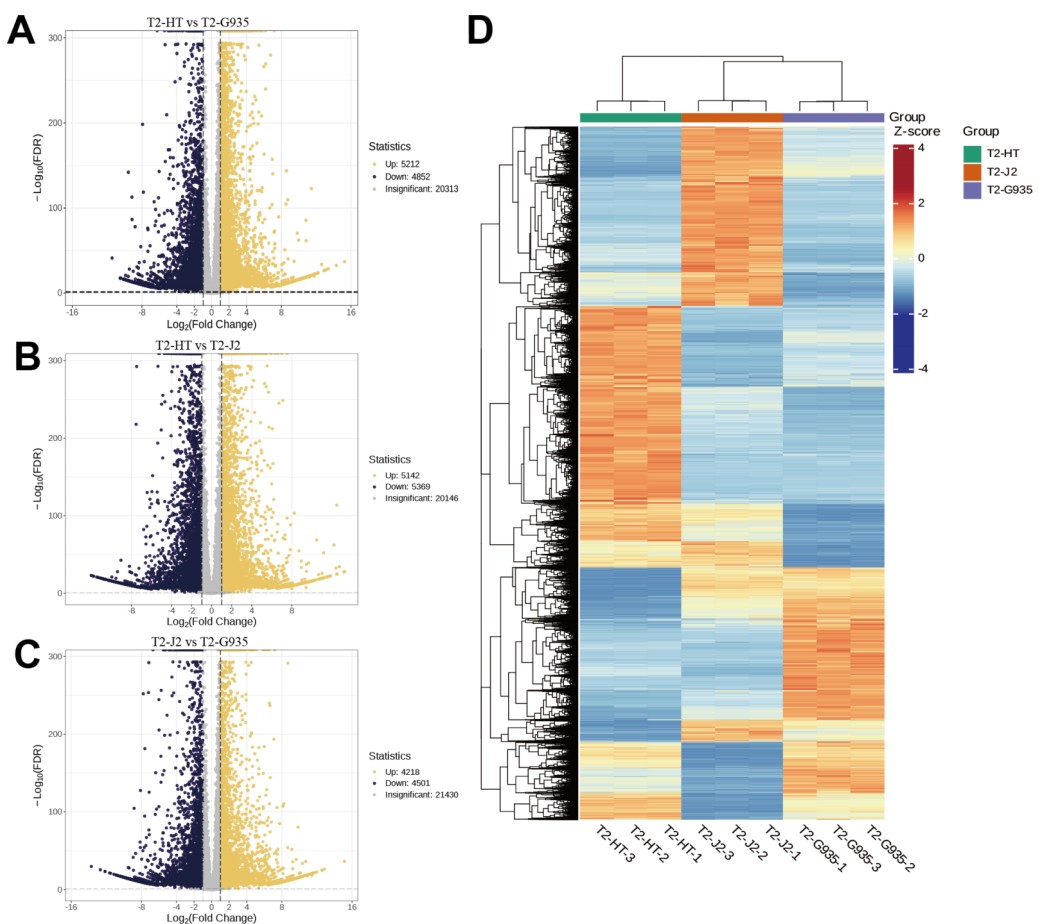

**Figure 5** **Volcano plots of DEGs and cluster heat map.** (A) Volcano plots of DEGs for T2/HT *vs.* T2/G935. (B) Volcano plots of DEGs for T2/HT *vs.* T2/J2. (C) Volcano plots of DEGs for T2/J2 *vs.* T2/G935. (D) Cluster heat map. Red color indicates high expression, and blue color indicates low expression.

which were involved in plant hormone signal transduction, plant-pathogen interaction and MAPK signaling pathway (Table S8). The key differential genes "*MD02G1013000*, *MD05G1229400*, *MD02G1149900*, *MD04G1238200*, *MD04G1238300*, *MD04G1238400*, *MD00G1016500*" were associated with root growth and development and were involved in plant hormone signal transduction pathways. The key differential genes "*MD02G1071000*, *MD02G1267200*, *MD05G1243600*, *MD02G1013000*, *MD05G1229400*, *MD02G1149900*, *MD04G1238200*, *MD04G1238300*, *MD04G1238400*, *MD02G1097900*" were associated with root growth and development and were involved in MAPK signaling pathways. The key differential genes "*MD04G1238200*, *MD04G1238300*, *MD04G1238400*, *MD02G1097900*, *MD02G1007200*, *MD02G1067500*, *MD03G1062400*, *MD03G1063500*, *MD03G1063600*, *MD04G1237900*, *MD04G1238100*, *MD04G1243600*, *MD06G1100500*, *MD09G1069300*, *MD09G1069400*" were associated with root growth and development and were involved in plant-pathogen interaction pathways.

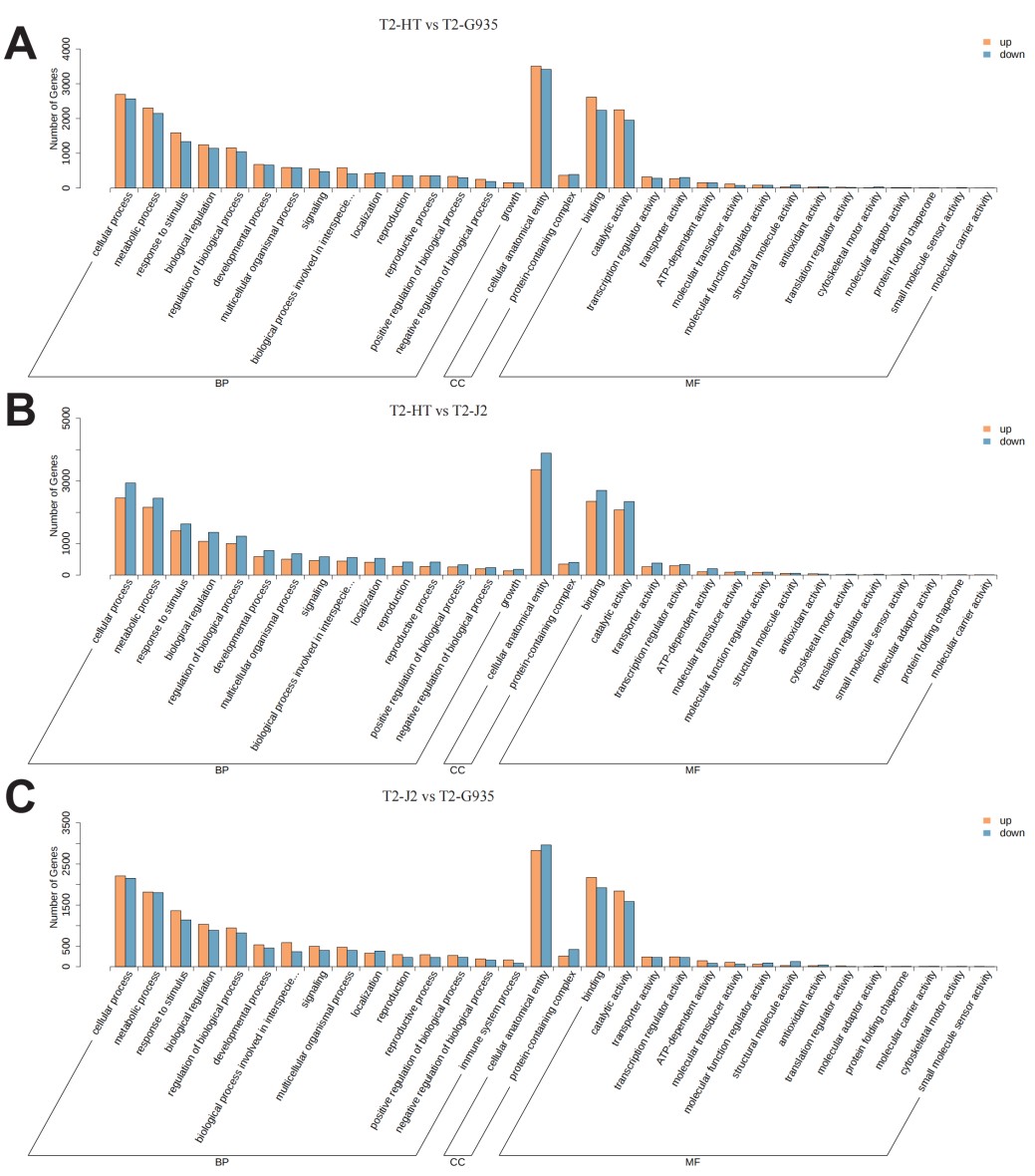

**Figure 6  Functional categorization of DEGs based on GO database.** (A) T2/HT *vs.* T2/G935. (B) T2/HT *vs.* T2/J2. (C) T2/J2 *vs.* T2/G935. The bar indicates gene numbers.

## Validation of DEGs by RT-qPCR

To confirm our transcriptome results, the relative expression of four DEGs was determined. The results of RT-qPCR and RNA-Seq were consistent for all four DEGs. These results indicate that reliable RNA-Seq data were obtained from the samples (Fig. S1).

## DISCUSSION

### Characteristics and impact of rootstocks

Modern fruit tree production requires the utilization of composite plants comprising both rootstock and scion components (*Zhao et al., 2023*). Rootstocks offer advantageous

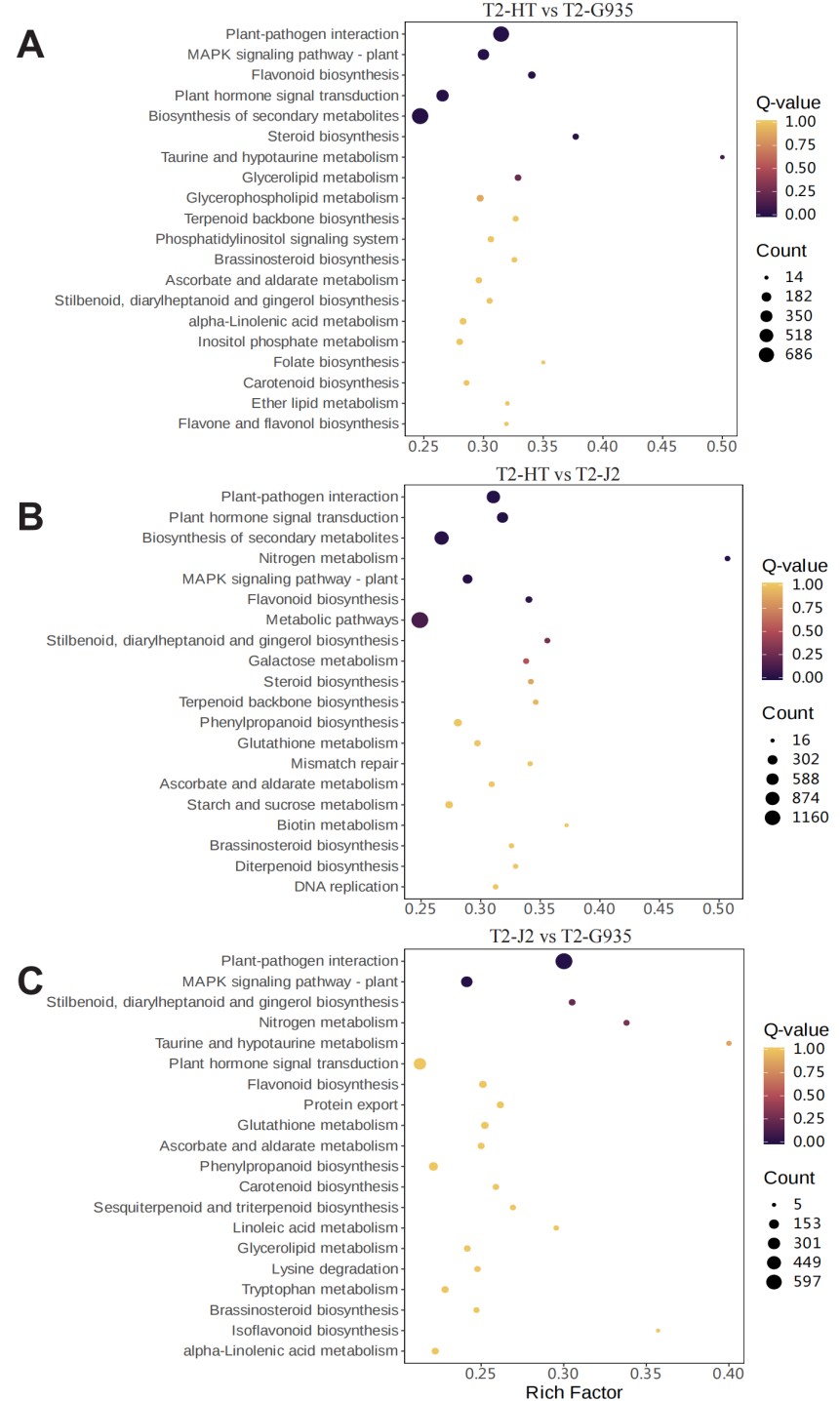

**Figure 7** **Enrichment scatter diagram.** (A) T2/HT *vs.* T2/G935. (B) T2/HT *vs.* T2/J2. (C) T2/J2 *vs.* T2/G935.

characteristics to plants, including resistance or resilience to abiotic or biotic stresses, as well as enhanced yield and fruit quality (*Vahdati et al., 2021*). Dwarfing rootstock can allocate more energy to the shoot, which improves fruit quality and yield (*Hayat et al., 2020*; *Opazo et al., 2020*). However, few studies have examined the mechanisms that control the effects of rootstocks on roots. For grafted fruit trees, the inherent stunting ability and growth and development characteristics of rootstocks lead to significant differences in tree root architecture (*Paltineanu et al., 2017*; *Zhang et al., 2021*). The root is an important organ of apple rootstock, especially dwarfing rootstock. When the same T2 scion was grafted to the vigorous rootstock HT and the dwarfing rootstocks J2 and G935, the root architecture indexes of the vigorous rootstock–scion combination (*i.e.,* root length, surface area, volume, forks, and tips) were better than those of the dwarfing rootstock–scion combination throughout the tree development period. Studies have shown that vigorous rootstocks have a greater root length density and more stratified vertical distribution than dwarfing rootstocks (*De Silva et al., 1999*). In addition, dwarfing rootstocks have a smaller fine root diameter, shorter life span, lower root length, and fewer root systems (*Hou et al., 2012*). The roots of dwarfing rootstocks are shallower in the soil than those of vigorous rootstocks (*An et al., 2017*). In addition, when the same scion was grafted to dwarfing rootstock, the growth vigor, leaf area, and photosynthetic capacity were reduced compared with the vigorous rootstock (*Zhou et al., 2020*). The root systems of apple rootstocks play a crucial role in regulating height of trees. (*Tworkoski & Miller, 2007*). In this study, the shoot lengths of HT and G935 were significantly higher than those of J2. A strong root system helps the root absorb minerals and water, which benefits the growth of apple plants (*Wu & Cheng, 2014*).

## Role of plant hormones in root development

Rootstocks affect the scion and root structure by changing the type and levels of plant hormones, including IAA, ABA, and $GA_3$ (*Van Hooijdonk et al., 2011*). In this study, we analyzed hormones in the roots and mature leaves of three rootstock varieties. Auxins, mainly IAA, are controlled by almost all aspects of plant development and have a decisive role in the root formation hormone network. In roots, IAA triggers lateral root morphogenesis and regulates cell elongation, root meristem activity, and root hair development (*Zeng et al., 2023*). Auxin plays a crucial role in regulating root growth and development, and its distribution pattern in plants significantly affects root plasticity (*Zhou et al., 2024*). A high IAA concentration enhanced root system growth, resulting in large root growth of the combination of vigorous rootstock and scion, and then promoted shoot growth, leading to robust overall tree development (*Seo et al., 2021*). Previous studies have shown that root formation is a complex biological process in apple rootstocks that is mainly influenced by auxin signaling pathways (*Tahir et al., 2021*). $GA_3$, being a crucial plant hormone, governs various aspects of and development across the plant's life cycle, encompassing cell elongation seed germination, and secondary growth (*Hong et al., 2021*). In the last decade, $GA_3$ has emerged as a key regulator of root meristem development (*Shtin, Dello Ioio & Del Bianco, 2022*). In this study, the root indices of the vigorous rootstock–scion combination were significantly higher than those of the dwarfing

rootstock–scion combinations (Fig. 2), and the GA$_3$ content in the roots of the vigorous rootstock–scion combination was also higher than that of the dwarfing rootstock–scion combinations (Fig. 3). ABA inhibits rooting and accumulates in plant varieties that are difficult to root (*Negishi et al., 2014*). In this study, the ABA content in the root system of the vigorous rootstock–scion combination was significantly lower than that of the dwarfing rootstock–scion combinations, which may be due to the high ABA concentration in the root system of the dwarfing rootstock–scion combinations inhibiting the development of lateral roots, making the root system growth weaker than that of the vigorous rootstock–scion combination (*Sun et al., 2018*).

## The role of different signaling pathways in root development of different rootstocks

Root formation in different rootstocks is a multi-step process that includes plant hormone signal transduction and the MAPK signaling pathway–plant (*Pi et al., 2023*). Root formation is regulated by the auxin pathway as well as several other hormone signaling pathways. Hormone signals are relevant for root development as they may be involved in adventitious root formation (*Tahir et al., 2022*). For example, plant root development under salt stress induced by phytohormone signal transduction pathway is resistant to salt stress (*Duan et al., 2022*). Differential genes in hormone signal transduction pathways are significantly enriched after exogenous application of auxin under potassium deficiency (*Zhou et al., 2024*). Transcriptional changes associated with phytohormone signaling in rice, which may be directly related to root phenotypic changes (*Zhang et al., 2023*). In the present study, KEGG analysis showed significant enrichment of plant hormone signaling genes in two comparisons, T2/HT *vs.* T2/G935 and T2/HT *vs.* T2/J2, confirming the importance of this pathway in root formation. In addition, KEGG analysis indicated significant enrichment of MAPK signaling pathway–plant and plant-pathogen interaction genes in the three comparison groups (Fig. 7). Previous studies have shown that MAPK and auxin signaling play positive roles in regulating root growth (*Zhao et al., 2015*). MAPK signaling pathway–plant has great potential to improve plant growth, playing an important role in stabilizing yield, producing robust root systems, and improving lodging resistance in plants (*Xiao et al., 2021*). In Cd-stressed rice, MAPKs are involved in root growth regulation *via* auxin signaling and modified expression of cell cycle genes (*Zhao et al., 2013*). There may also be competitive interactions between various pathogens that infect apple roots and affect apple root development (*Emmett et al., 2014*). By influencing the spatiotemporal availability of susceptible tissues, root development can govern plant-pathogen interactions with soil-borne pathogens, with less susceptible rootstocks maintaining root growth rates in replanted soil, as compared to susceptible rootstocks (*Henfrey, Baab & Schmitz, 2015*). Some key differential genes related to root growth and development were involved in plant hormone signal transduction, plant-pathogen interaction and MAPK signaling pathway, which may play important roles in differences in the rooting ability of different rootstocks. These results suggest that the stronger rooting ability of HT and G935 is related to these pathways.

## CONCLUSIONS

In this study, three rootstock/scion combinations were found to have different rooting abilities. The molecular mechanism of the rooting differences among the rootstocks was investigated further. The results showed that the mature leaves and roots of HT had higher IAA and GA$_3$ levels, and lower ABA levels. Transcriptome analysis revealed that several key differential genes associated with root growth and development were involved in plant hormone signaling, MAPK signaling pathway-plant, and plant-pathogen interaction, and play important roles in the differences in the rooting ability of different rootstocks.

### Funding

This work was supported by the Key Research and Development Project of Hebei Province (20326802D), the Apple Innovation Team of Modern Agricultural Industrial Technology System of Hebei Province (HBCT2023120403), and the earmarked fund for the Agriculture Research System of China (CARS-27). The funders had no role in study design, data collection and analysis, decision to publish, or preparation of the manuscript.

### Grant Disclosures

The following grant information was disclosed by the authors:
Key Research and Development Project of Hebei Province: 20326802D.
Apple Innovation Team of Modern Agricultural Industrial Technology System of Hebei Province: HBCT2023120403.
Agriculture Research System of China: CARS-27.

### Competing Interests

The authors declare there are no competing interests.

### Author Contributions

- Zhongyong Li conceived and designed the experiments, authored or reviewed drafts of the article, and approved the final draft.
- Yang Cao conceived and designed the experiments, performed the experiments, analyzed the data, authored or reviewed drafts of the article, and approved the final draft.
- Jie Zhu performed the experiments, analyzed the data, authored or reviewed drafts of the article, and approved the final draft.
- Jin Liu analyzed the data, authored or reviewed drafts of the article, and approved the final draft.
- Feng Li analyzed the data, prepared figures and/or tables, and approved the final draft.
- Shasha Zhou analyzed the data, prepared figures and/or tables, and approved the final draft.
- Xueying Zhang analyzed the data, prepared figures and/or tables, and approved the final draft.

- Jizhong Xu conceived and designed the experiments, prepared figures and/or tables, and approved the final draft.
- Bowen Liang conceived and designed the experiments, authored or reviewed drafts of the article, and approved the final draft.

## Data Availability

The raw sequencing data are available at the NCBI Sequence Read Archive: PRJNA1066895.

The raw data are available in the Supplemental File.

## Supplemental Information

Supplemental information for this article can be found online at http://dx.doi.org/10.7717/peerj.18244#supplemental-information.

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
