# Peer review of "Comparative transcriptome and hormone analyses of roots in apple among three rootstocks with different rooting abilities"

_PeerJ, doi:10.7717/peerj.18244_

## Round 0.1 · original submission · Major Revisions

Please address issues pointed by the reviewers and revise manuscript accordingly.

Reviewer 1 ·

Basic reporting

The research article “Comparative transcriptome and hormone analyses of roots in apple among three rootstocks with different rooting abilities” aligns with the journal’s vision and Scope. The research focuses on comparing the effects of Malus robusta (HT)/Tianhong 2 (T2), G935/T2, and Jizhen 2 (J2)/T2 combinations. The findings significantly contribute to understanding the genetic and hormonal factors that underpin root development in fruit trees.
However, I recommend some major additions before acceptance of the manuscript.
1. The authors need to report some key genes responsible for this specific mechanism.
2. Line 37, apple (Malus domestica) scientific name must be in italic style.
3. Highlight the importance of grafting in detail and then narrow down to scion/rootstock interactions.
4. On line 68, the correct representation of gibberellic acid should be in subscript as GA₃.
5. The authors need to revise the introduction section with recent literature.
6. Lines 86-87 should include comprehensive details about the growing conditions in the greenhouse used for this experiment i.e., temperature, light, fertilizers, irrigation plan, etc.
7. The manuscript should include a clear description of the grafting method and timing used in the experiment.
8. The manuscript should clarify that the sampling points in Figure 1 correspond to 0, 30, 60, 90, and 120 days after grafting, detailing when each sample was collected and the same for the rest of the figures.
9. The authors should address the reasons behind not measuring cytokinin levels, given the study’s focus on hormone and transcriptome comparisons among various apple rootstocks.
10. Why authors did not focus on DEGs Related to Plant Hormone Signaling Transduction Pathway. I did not see hormone signaling pathways or key genes. Add in the revised manuscript.
11. The manuscript should include detailed plant images at each sampling point to visually illustrate the differences observed. Adding these figures will help readers better understand the variations in plant responses across different time points.

I recommend major revisions.

Experimental design

experimental design is reasonable.

Validity of the findings

need to revised significantly, must add some hormone signaling pathways etc and report some key genes.

Annotated reviews are not available for download in order to protect the identity of reviewers who chose to remain anonymous.

·

Basic reporting

The article “Comparative transcriptome and hormone analyses of roots in apple among three rootstocks with different rooting abilities” falls within the aims and scope of the journal.

Experimental design

The research design was appropriate.

Validity of the findings

no comment

Additional comments

Dear the authors,
I have directly commented and marked the comments on the manuscript under the file name: (peerj-101320-manuscript_Peer-review).
More detail for the places in the text that need to be corrected and revised had been commented directly in the manuscript. The manuscript should be represented in the section “Results” and some in the section “Discussion”.
Please use it for easy tracking and revising.

Reviewer 3 ·

Basic reporting

The English language has major drawbacks that should be improved professionally.
There are several wrong citations in the manuscript

Experimental design

Experimental design is reasonable. However, they roughly analyzed the RNA sequencing. I am not satisfied with their analysis

Validity of the findings

They did not analyze the RNA seq data properly. Therefore, the findings are not robust

Additional comments

Line 20–21: Apple root architecture plays an important role in the growth and development of fruit trees…. Apple roots did not play any role in the growth of other fruit trees.
Line 37: Malus domestica…. should be written in italics.
What are the main problems that apple trees faced during the development of the rooting system?
Line 71: increaseali…typos
Line 85: How many seedlings were used in this study?
Line 92: What do you mean by destructive sampling?
Line 99: Sun el al., 2021…..Line 106 Han et al., 2022…..Line 109: Guo et a;., 2022. Why have you cited these studies here? There is no relation between these and the text. There are several wrong citations in the above text, including the introduction. Delete all the irrelevant citations from the whole text.
Line 110: qRT-PCR, its RT-qPCR. Line 111.
Line 111: This line has no clear meaning. Rewrite it
Line 112: Provide the gene names.
Line 159: PCA, provide the complete name.
I did not know why the authors did RNA-seq analysis. They did not identify any gene responses. In this way, RNA sequencing is totally useless. Please read the articles provided below to make your study clearer and more useful.
Tahir M M, Chen S, Ma X, Li S, Zhang X, Shao Y, Shalmani A, Zhao C, Bao L, Zhang D. 2021a. Transcriptome analysis reveals the promotive effect of potassium by hormones and sugar signaling pathways during adventitious roots formation in the apple rootstock. Plant Physiology and Biochemistry, 165, 123-136.
Tahir M M, Tong L, Fan L, Liu Z, Li S, Zhang X, Li K, Shao Y, Zhang D, Mao J. 2022c. Insights into the complicated networks contribute to adventitious rooting in transgenic MdWOX11 apple microshoots under nitrate treatments. Plant, Cell & Environment, 45, 3134-3156.
The discussion part also improves after analyzing the RNA seq data correctly.
The English language has major drawbacks that should be improved professionally.

---

## Round 0.2 · Minor Revisions

Please address the remaining concerns of the reviewers and revise the manuscript accordingly.

Reviewer 1 ·

Basic reporting

Literature needs to be updated.
Authors should must report some key genes responsible for this specific mechanims.

Experimental design

reasonable

Validity of the findings

Have novality

Additional comments

Please accept it with minor corrections.
The authors have made good efforts to revise the manuscript. However, the literature review would benefit from incorporating more recent findings to ensure it reflects the latest advancements.
I recommended some minor revisions.
1. Additionally, the authors must highlight key genes involved in the specific mechanisms discussed.
2. The discussion section should be refined to provide more focused and in-depth insights into the subject matter.
3. You may consider incorporating recent examples from the literature to strengthen your manuscript.
4. Please improve the abstract and conclusion parts.
5. Focus on hormone-related pathways and report some key genes responsible for different rooting abilities

·

Basic reporting

Dear authors,
I have directly commented and marked the comments on the manuscript (.docx file) with a total of 29 comments [Commented GDV1-29]. However, the authors have just picked up 4 comments (GDV15, 17, 22, and 29) to answer and report in the cover letter, file named “peerj-101320-R1_Author_Comments”.
Therefore, I had to review the revised manuscript to ensure my comments have been addressed. Fortunately, all my concerns have been resolved.
In the future, the authors should bring up all comments and responses point by point and show them in the cover letter.
All my concerns have been addressed. I have no more comments.
Thank you for your hard work and effort in revising the manuscript.

Experimental design

no comment

Validity of the findings

no comment

Additional comments

no comment

Reviewer 3 ·

Basic reporting

Good

Experimental design

Good

Validity of the findings

The authors did not identify key genes; the present form of the manuscript is very basic. I suggest identifying DEGs related to the plant hormone signaling transduction pathway.

Additional comments

Line 283: adventitic ......typo

---

## Round 0.3 · accepted · Accept

All remaining issues pointed out by the reviewer were addressed and the revised manuscript is acceptable now.